

# *In vitro* cytoprotective and *in vivo* anti-oral mucositis effects of melatonin and its derivatives

Pramote Mahakunakorn[1], Pimpichaya Sangchart[2], Panyada Panyatip[3], Juthamat Ratha[4], Teerasak Damrongrungruang[5], Aroonsri Priprem[4,6] and Ploenthip Puthongking[4,7]

[1] Division of Pharmacognosy and Toxicology, Faculty of Pharmaceutical Sciences, Khon Kaen University, Khon Kaen, Thailand
[2] Department of Pharmaceutical Technology, Faculty of Pharmacy, Srinakharinwirot University, Nakhon Nayok, Thailand
[3] Department of Pharmacognosy, Faculty of Pharmacy, Srinakharinwirot University, Nakhon Nayok, Thailand
[4] Melatonin Research Group, Khon Kaen University, Khon Kaen, Thailand
[5] Department of Oral Biomedical Sciences, Faculty of Dentistry, Khon Kaen University, Khon Kaen, Thailand
[6] Faculty of Pharmacy, Mahasarakham University, Mahasarakham, Thailand
[7] Division of Pharmaceutical Chemistry, Faculty of Pharmaceutical Sciences, Khon Kaen University, Khon Kaen, Thailand

Corresponding author
Ploenthip Puthongking,
pploenthip@kku.ac.th

## ABSTRACT

According to our preliminary study, melatonin and its *N*-amide derivatives (*N*-(2-(1-4-bromobenzoyl-5-methoxy-1H-indol-3-yl)ethyl)acetamide (BBM) and 4-bromo-*N*-(2-(5-methoxy-1H-indol-3-yl)ethyl)benzamide (EBM)) inhibited the marker of acute inflammation in tests *in vitro* and *in vivo*. The anti-inflammatory agent is intended for the prevention and treatment of chemotherapy-induced toxicity. In this study aimed to evaluate the effect of melatonin and its derivatives on mechanisms related to chemotherapy-induced oral mucositis by *in vitro* ROS and 5-FU-induced human keratinocyte cells as well as *in vivo* oral mucositis model. In *in vitro* $H_2O_2$-induced HaCaT cells, BBM had the highest level of protection (34.57%) at a concentration 50 μM, followed by EBM (26.41%), and melatonin (7.9%). BBM also protected cells against 5-FU-induced to 37.69–27.25% at 12.5–100 μM while EBM was 36.93–29.33% and melatonin was 22.5–11.39%. In *in vivo* 5-FU-induced oral mucositis in mice, melatonin, BBM, and EBM gel formulations protected tissue damage from 5-FU similar to the standard compound, benzydamine. Moreover, the weight of mice and food consumption recovered more quickly in the BBM group. These findings suggested that it was possible to develop BBM and EBM as new therapeutic agents for the treatment of oral mucositis.

## INTRODUCTION

Oral mucositis is a distressing adverse effect of 5-fluorouracil (5-FU) (*Plevová, 1999*), a pyrimidine analogue that inhibits RNA and DNA synthesis, extensively used in the

**Figure 1** **Structure of melatonin and its derivatives.**

treatment of cancer (*Focaccetti et al., 2015*). In 5-FU-treated patients, mucositis, inflammation of the mucosal membrane that occurs throughout the gastrointestinal tract (GIT), is mostly found in the intestine (*Sonis, 2004*). The mechanism of oral mucositis is explained as five phases including initiation, primary damage response, signal amplification, ulceration, and healing. This mechanism is associated with inflammation, generation of reactive oxygen species (ROS), and wound healing (*Sonis, 2007*). Agents with anti-inflammatory and antioxidant activities are purposed strategies for preventing and treating the chemotherapy-induced toxicity.

Melatonin, an endogenous hormone mainly produced and secreted by the pineal gland, has been reported to have various biological activities such as antioxidant (*Bonnefont-Rousselot & Collin, 2010*), immunomodulation (*Srinivasan et al., 2008*), and anti-inflammatory effect (*Mayo et al., 2005*). However, melatonin has low oral bioavailability, typically ranging from 9 to 33% (*Harpsøe et al., 2015*), and a short half-life as it is metabolized by the CYP1A2 enzyme (*Waldhauser et al., 1984*; *Tordjman et al., 2013*). *N*-amide melatonin derivatives that mimic the 4-chlorobenzoyl residue of indomethacin (non-steroidal anti-inflammatory drugs; NSAIDs), were synthesized by our group (Fig. 1). Derivatives with bromobenzoylamide-substitutions showed better antioxidant ability than unsubstituted derivatives, as evidenced by changes in the electron spin resonance (ESR) signal (*Panyatip, Nunthaboot & Puthongking, 2020*). This result is supported by the study of *Mor et al. (2004)*, which confirmed that increasing lipophilicity at the indole ring of melatonin improved antioxidant activity. Interestingly, an *in silico* study predicted that the *N*1-position lipophilic-substituted melatonin derivative BBM (*N*-(2-(1-4-bromobenzoyl-5-methoxy-1H-indol-3-yl)ethyl) acetamide) could not be metabolized by CYP1A2, prolonging its half-life compared to melatonin (*Panyatip, Nunthaboot & Puthongking, 2020*). Another melatonin derivative with a

4-bromobenzoyl amide substitution at the *N*-acyl side chain EBM (4-bromo-*N*-(2-(5-methoxy-1H-indol-3-yl)ethyl) benzamide) displayed potent antioxidant and neuroprotective effects, but was predicted to be a substrate of CYP1A2 (*Panyatip et al., 2020a*, *2020b*). Addition, BBM and EBM exhibited superior inhibition of the production of NO, PGE2 and IL-6 in LPS-induced inflammatory responses in RAW 264.7 macrophage and reduced mice ear edema induced by croton oil (*Sangchart et al., 2021*). Based on their antioxidant and anti-inflammatory properties, this study aimed to investigate the effects of melatonin derivatives as chemo preventive agents against $H_2O_2$ and 5-FU-induced cellular toxicity using an *in vitro* oral mucositis model based on HaCaT keratinocytes and an *in vivo* oral mucositis model in mice (*Shimamura et al., 2018*; *dos Santos Filho et al., 2018*).

# MATERIALS AND METHODS

## *In vitro* studies on HaCaT cells

The immortalized human keratinocyte (HaCaT) cell line (American Type Culture Collection, Manassas, VA, USA) were prepared by seeding into 96-well plates to obtain a density of about $2 \times 10^4$ cells/well before use, cultured in 100 µl Dulbecco's modified Minimum Essential medium (DMEM) containing 4.5 g/l D-glucose, L-glutamine, 1% non-essential amino acid (NEAA) and 110 mg/l sodium pyruvate. To assay cytotoxicity, 100 µl of melatonin in DMEM (final concentration of 12.5–2,000 µM), BBM or EBM (final concentration 12.5–100 µM) were treated to the cells and incubated for 24 h. MTT assay was used to determine by incubation the 3-(4, 5-dimethyl-2-thiazolyl)-2, 5-diphenyl-2H-tetrazolium bromide (MTT, Invitrogen, Waltham, MA, USA). A total of 50 µl of MTT (final concentration 0.5 mg/ml) was incubated to cells for 4-h, followed by removal of the cultured medium and then adding 0.1 ml of dimethyl sulfoxide (DMSO, Fisher Chemical, Loughborough, UK) to dissolve formazan which was quantitatively analyzed by measuring UV absorbance at 550 nm using a microplate reader (EnsightTM, Perkin Elmer Inc., Waltham, MA, USA). Cell viability was calculated as follows;

$$\% \text{cell viability} = \frac{\text{Absorbance of Sample}}{\text{Absorbance of Control}} \times 100.$$

## Cytoprotective effect in $H_2O_2$-damaged cells

$H_2O_2$ (Invitrogen, Eugene, Oregon, USA), 0.1–2 mM, was co-cultured with HaCaT cells, the viability of which were determined to select its concentration for use. The prepared cells were treated with 12.5–100 µM of melatonin or BBM and EBM, for 3 h, followed by 24-h incubation with the selected concentration of $H_2O_2$. The protective effect of melatonin or BBM or EBM against $H_2O_2$-damaged cells was calculated. The percentage of cell protective was calculated as follows;

$$\% \text{ protective effect} = \frac{\text{no. of cell death of } H_2O_2 \text{ treated } - \text{ no. of cell death of sample} + H_2O_2 \text{ treated}}{\text{no. of cell death of } H_2O_2} \times 100$$

where, cell death being estimated from 100–%cell viability. The $H_2O_2$ group was considered as 0% cell protective.

## Cellular antioxidant activity

Cellular antioxidant activity assay was performed to evaluate the activity of an antioxidant compounds that prevent the oxidation of 2′,7′-dichlorodihydrofluorescein (DCFH) and inhibit the formation of fluorescent 2′,7′-dichlorofluorescein (DCF). The cultured medium of HaCaT cells, seeded to a 96-well fluorescent plate ($2 \times 10^4$ cells/well, 100 μl) and incubated for 24 h, was removed and replaced with serum-free medium. The cells were then treated with 100 μl of melatonin or BBM, and EBM and subsequently incubated with 25 μM of DCFH-DA (2′,7′-dichlorodihydrofluorescein diacetate; Sigma-Aldrich, St. Louis, MO, USA) in the medium for 1 h. After washing by phosphate buffer solution (PBS, Gibco. Inc., New York, NY, USA), the cells were reacted with 0.1 ml 2,2′-azobis (2-methylpropionamidine) dihydrochloride (AAPH, Sigma-Aldrich, St. Louis, MO, USA) (0.6 mM) in Hank's balanced salt solution (HBSS, Gibco. Inc., New York, NY, USA) (AAPH/HBSS) and subjected to 5-min repeated measurements of fluorescent intensities, $\lambda_{exc}$ and $\lambda_{ems}$ of 485 and 538 nm, respectively, for 1 h at a controlled temperature of 37 °C using a microplate reader (SpectraMax Gemini EM, Molecular Devices Microplate Reader, San Jose, CA, USA). The negative control was vehicle and HBSS-treated cells, while the positive control being vehicle and AAPH/HBSS-treated cells. Area under the fluorescent curve (AUC) *vs* time was integrated by the software (SpectraMax Gemini EM, Molecular Devices Microplate Reader, San Jose, CA, USA) and quantified as cellular antioxidant activity as quercetin-equivalence using a standard curve of quercetin (Sigma-Aldrich, St. Louis, MO, USA), dissolved in DMSO, 10–100 μM, and reacted with AAPH/HBSS before measurements of fluorescence intensities, followed by linear regression analysis.

## Chemoprotective effect against 5-fluorouracil induced cells damage

To titrate the concentration of 5-FU on HaCaT cells, the above density of prepared cells was 24-h treated with 100 μl of 5-FU in medium (final concentration of 0.01–0.31 mM), followed by MTT assay as previously described.

The concentration of 5-FU of 0.02 mM, was selected for use as it has been reported (*dos Santos Filho et al., 2018*; *Harada et al., 2018*). Chemoprotective effect of melatonin, BBM or EBM on HaCaT cells were investigated after incubating 12.5–100 μM of melatonin or BBM or EBM with the prepared cells for 3 h, then with 5-FU for 24 h and subjected to the described MTT. Chemoprotective effect (%) of melatonin was calculated as follows;

$$\% \text{ chemoprotective effect } = \frac{\text{cell death of 5-FU treated} - \text{cell death of melatoin} + \text{5-FU treated}}{\text{cell death of 5-FU treated}} \times 100$$

Similarly, chemoprotective effects of BBM and EBM were also estimated. Intercellular reactive oxygen species (ROS) of the prepared cells treated with melatonin or BBM or EBM and 5-FU were assayed. After treatments, the cells were washed with PBS, incubated with 100 μl of DCFH-DA in medium (25 μM) for 1 h, washed with PBS and determined for

fluorescence intensities using the previously described method. Results of the 5-FU-treated cells represented the control group, *i.e.* 100% ROS level, and those of the samples were calculated in relative to the control group, as follows;

$$\% \text{ ROS level of the sample } = \frac{\text{fluorescent intensity of sample}}{\text{fluorescent intensity of control}} \times 100.$$

### *In vivo* 5-FU-induced oral mucositis in mice

An aqueous gel base was modified from *Priprem et al. (2018)*, composed of 10%w/w of poloxamer 407 (P407; BASF, Oakville, ON, Canada), 1%w/w of hydroxypropyl methylcellulose (HPMC; Samsung Fine Chemicals, Korea), and 10%w/w of polyvinylpyrrolidone K90 (PVP; Dai-ichi Kogyo Seiyaku, Japan), was thoroughly mixed. 1%w/v of melatonin, BBM, EBM or benzydamine (Sigma-Aldrich, St. Louis, MO, USA) which was dispersed in 20%w/w of polyethylene glycol 400 (PEG, Sigma-Aldrich, St. Louis, MO, USA) before thoroughly mixing with the gel base.

Prior the study, the protocol was approved by the Institutional Animal Ethics Committee (IACUC-KKU 79/60), Khon Kaen University, Thailand, and handled in accordance to expectations for animal care. Male ICR mice 6–8 week-old ($n = 40$; Nomura Siam International Co, Ltd., Bangkok, Thailand) were 1 week acclimated and maintained to a 12 h light/dark cycle, controlled at $23 \pm 2\,°C/50$ –70% relative humidity and ventilated, at the Northeast Laboratory Animal Center, Khon Kaen University, Thailand. Animals were housed in groups in the 26.6 cm × 42.5 cm × 18.5 cm cage with wood-shaving bedding and provided *ad libitum* access to water and feed. All animals were housed socially in groups of six per cage. The bedding of all cages was changed every 2–3 days.

Mice were induced to oral mucositis by intraperitoneal injections with 5FU to evaluate the treatment effectiveness of the melatonin derivatives. The melatonin derivatives gels were applied to the oral cavity of the animals after inducing oral mucositis. This method of administration did not cause the animal pain thus analgesia was not used. During the experiments, the weights of mice and food consumed were recorded daily. If it was found that the animal had an allergic reaction to the substances, showing symptoms such as being unable to move or consume the food, reducing the weight of the animal by 15–20% within 2–3 days, or showing pain or suffering from the experiments, the animal would be separated and euthanized by the pentobarbital sodium (100 mg/kg) intraperitoneal injection. In this experiment, there is no eligible animal was euthanized prior to the planned end. After experiments, the animals were euthanized by intraperitoneal injection of pentobarbital sodium (100 mg/kg) to collect the tongue.

Mice (an average weight of $36.1 \pm 2.0$ g) were randomly assigned into six groups, *i.e.* a control group (no treatment, $n = 4$), 5FU/blank gel, 5FU/melatonin, 5FU/BBM, 5FU/EBM and 5FU/benzydamine ($n = 6$, each). 5FU 10 mg/ml in normal saline solution was injected (50 mg/kg) *via* intraperitoneal on D-6, D-4 and D-2, scheduled as shown in Fig. 2. The 0.05 ml of each gel sample was dropped daily into the oral cavity of the relevant mouse, from D0 until sacrifice on D6. One mouse in the 5FU/melatonin group died during the experimental period. Animals were euthanized and tongue samples were collected.

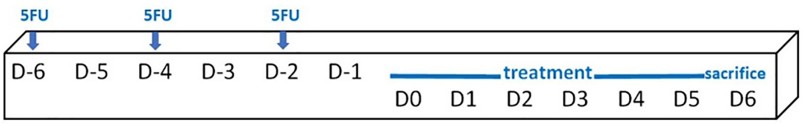

**Figure 2 Schedule of the *in vivo* study protocol.**

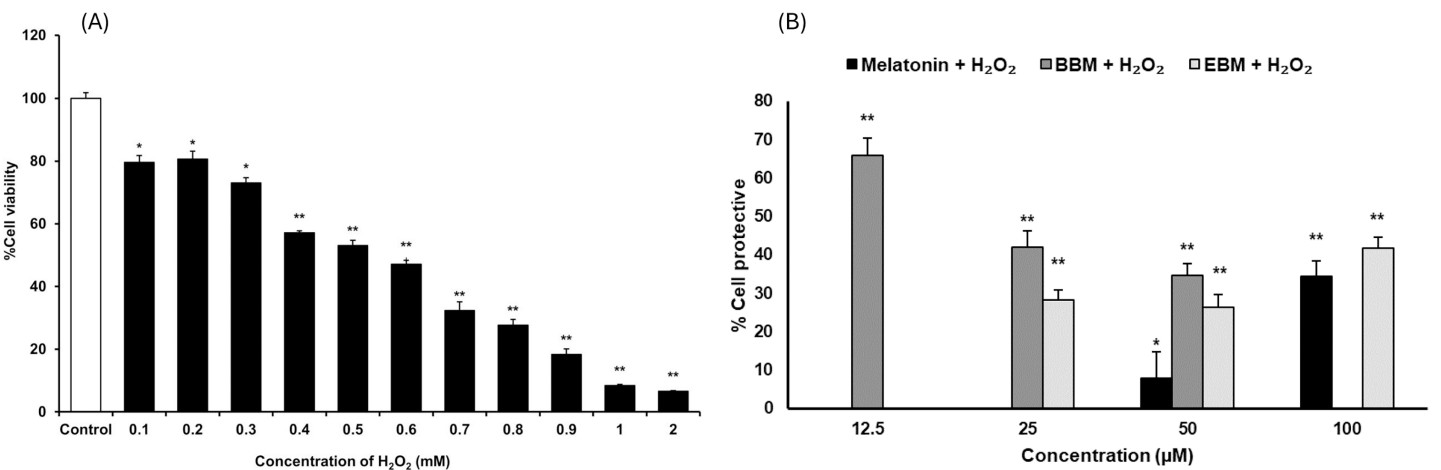

**Figure 3 Cytoprotective effects in $H_2O_2$-damaged HaCaT cells.** Values are expressed as mean ± SE ($n = 8$). $^*p < 0.05$, $^{**}p < 0.001$ *vs* the control group ($H_2O_2$ untreated group). (A) Effect of $H_2O_2$ on viability. (B) Cytoprotective effect of melatonin (black columns), BBM (grey columns) and EBM (stripes columns).

The tongue tissues, fixed in 4% (v/v) paraformaldehyde solution for 48 h and paraffin embedded, excised (5 μm-thickness) and stained with hematoxylin and eosin (H&E), were histologically evaluated, and measured keratinized and epithelial thicknesses using the inverted microscope (AE2000; Motic Incoporation, Ltd., Causeway Bay, HongKong) (20× magnification).

## Statistical analysis

All data are expressed as mean ± SD. Statistical analysis of the experimental data was carried out using the SPSS Statistic 21.0 program. The mean and standard deviation were compared using one-way analysis of variance (ANOVA) followed by Tukey's *post hoc* test. The criterion for statistical significance was $p < 0.05$.

## RESULTS

The cytotoxicity of melatonin, BBM, and EBM was investigated in the HaCaT cells. Melatonin was incubated with the cells at concentrations ranging from 12.5- to 2,000 μM, while BBM and EBM were used at concentrations ranging from 12.5 to 100 μM based on their solubility. Results obtained showed that cell viability was higher than 80% across all concentrations (Supplemental Data of Cytotoxic MLT EBM BBM). Thus, there was no cytotoxicity observed for any of the all compounds used in the present study.

As shown in Fig. 3A, the result illustrates incremental reductions in cell viability with increasing concentrations of $H_2O_2$. HaCaT cells exposed to 0.4 mM $H_2O_2$ exhibited an average viability of about 53%, indicating an optimal concentration for cytoprotective

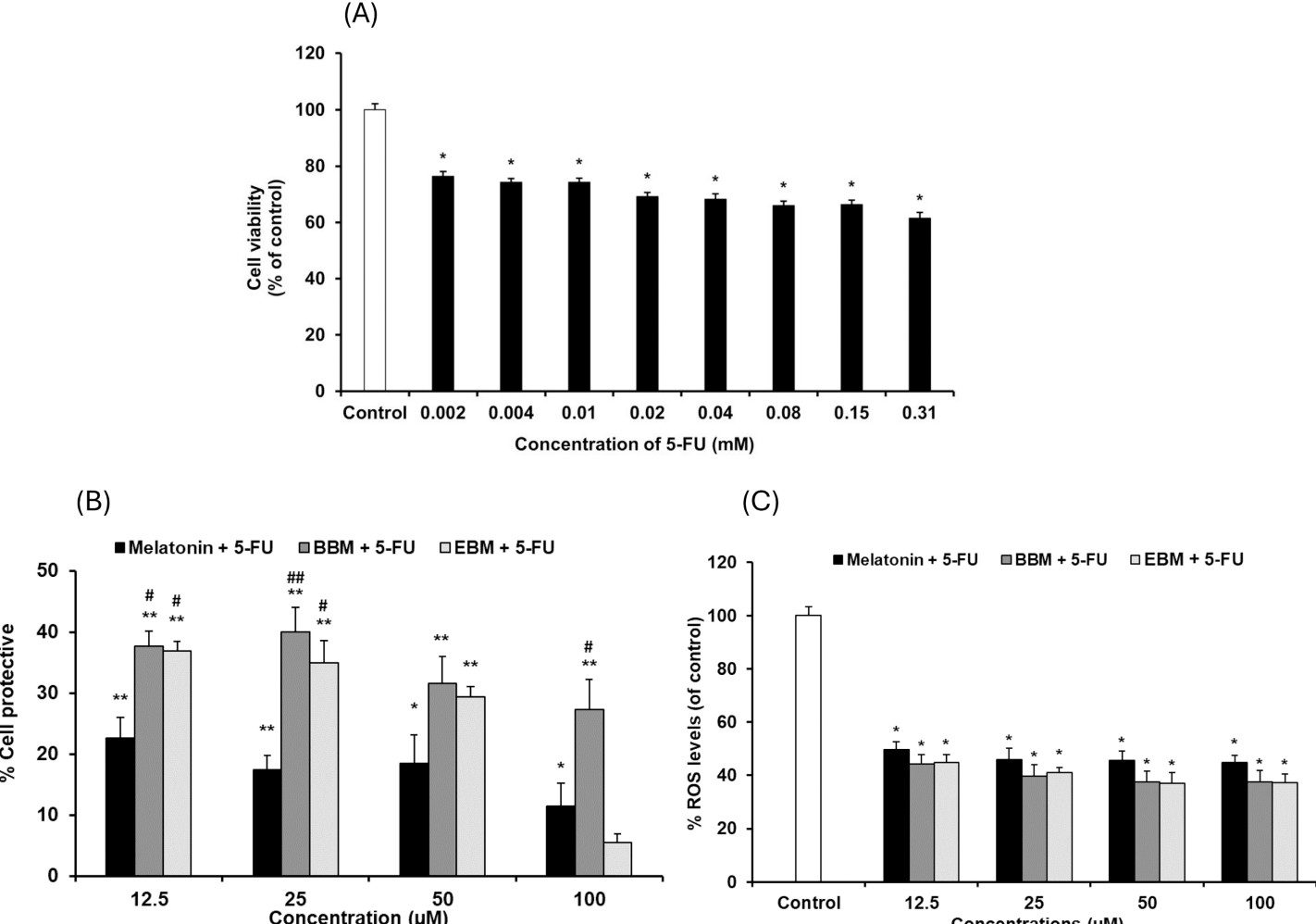

**Figure 4 Effect of 5-FU on HaCaT cells.** (A) Effect of 5-FU on cell viability, (B) Chemoprotective effect of melatonin (black columns), BBM (grey columns) and EBM (stripes columns) on HaCaT cells exposed to 0.02 mM of 5-FU, (C) ROS levels induced by 5-FU and co-cultured with various concentrations of melatonin, BBM and EBM. Values are expressed as mean ± SE. $^*p < 0.05$ vs control (5-FU treated group) ($n = 8$), $^{**}p < 0.001$ vs control, $^{\#}p < 0.05$, $^{\#\#}p < 0.001$ vs melatonin.

investigation. Figure 3B compares the cytoprotective effects against $H_2O_2$-damaged HaCaT cells, pretreated with melatonin, BBM, and EBM (12.5, 25, 50, 100 μM). Cells pretreated with 50 and 100 μM of melatonin showed 7.9 ± 6.8% and 34.5 ± 4.0% protection, while those treated with 12.5, 25, 50 μM of BBM showed 65.8 ± 4.5%, 42.0 ± 4.2% and 34.6 ± 3.2% protection, respectively. Similarly, cells pretreated with 25, 50, and 100 μM of EBM showed 28.2 ± 2.7%, 26.4 ± 3.2%, and 41.7 ± 2.9% protection, respectively.

The cytotoxicity of 5-FU-damaged HaCaT cells was determined by MTT assay, Fig. 4A shows significant reduction in cell viabilities (ranging between 62 and 76%) within the investigated concentration range, indicating a concentration-dependent effect. A concentration of 0.02 mM (2.5 μg/ml) of 5-FU was chosen for investigating the chemopreventive effects of the test compounds, as it resulted in partial viabilities of ≤70% for comparison of cell recovery. Figure 4B illustrates an inverse trend in cell protection

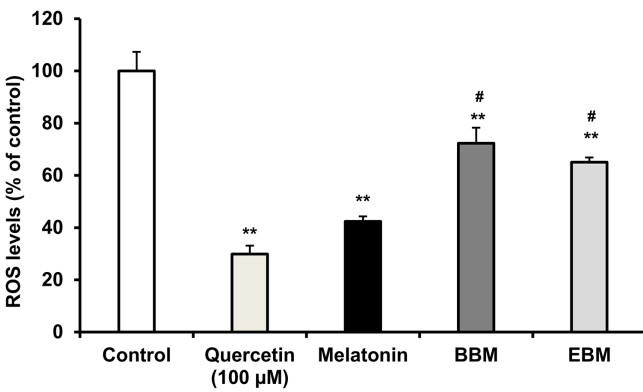

**Figure 5 ROS levels (% of control) in HaCaT cells by AAPH assay induced by AAPH and co-cultured with 50 μM of melatonin (black columns), BBM (grey columns) or EBM (stripes columns), quercetin (dots columns) and control (white columns) is AAPH treated group ($n = 8$).** Values are expressed as mean ± SE. $^{**}p < 0.001$ *vs* control, $^{#}p < 0.05$ *vs* melatonin. Control was AAPH treated group.

relative to the 5-FU treated group (control) with increasing concentrations of melatonin, BBM or EBM. Within the concentration range of 12.5–100 μM, melatonin provided chemoprotection ranging from 22.5 ± 3.4% to 1.4 ± 3.9%, while BBM provided chemoprotection ranging from 40.0 ± 4.0% to 27.3 ± 5.0% and EBM provided chemoprotection ranging from 36.9 ± 1.5% to 5.4 ± 1.5%. Figure 4C illustrates ROS levels from 5-FU-induced cells after treatment with melatonin or its derivatives. Results showed that ROS production was significantly suppressed after treatment with melatonin and its derivatives, with the levels of ROS tending to reduce with an increase in the concentration of the compounds. Both melatonin derivatives exhibited lower levels of ROS than melatonin. Taken together, melatonin derivatives could protect cells from 5-FU-induced damage, partly due to the suppression of ROS levels, leading to cell survival.

Figure 5 compares the ROS levels in HaCaT cells co-cultured with 50 μM of melatonin, BBM or EBM using an AAPH assay which depended on cellular uptake ability of the test compounds to exert inhibition on peroxyl radicals. The lowest ROS level of 30.0 ± 1.1% was obtained from 100 μM-quercetin co-cultured cells, estimating the antioxidant capacities of melatonin, BBM and EBM as quercetin equivalent at 80.6 ± 1.5, 35.6 ± 1.4 and 46.5 ± 4.5 μM, respectively.

The result from Fig. 6A illustrates initial increases in body weights of 5-FU-treated mice from D-6 until D-4, after the second dose of 5-FU, followed by gradual decreases of 4 to 6% weight loss on D-2 after receiving the second doses, with the lowest observed on D0. Diarrhea and hair loss were also noted since D-2. In contrary, the average body weights of mice in the normal group increased linearly at a rate of 0.8%/day (a correlation coefficient of 0.977) from D2–D7 without any adverse effects. In comparison to initial body weights (*i.e.*, D0) during the treatment duration (D2–D7) benzydamine, BBM, EBM, melatonin and placebo treated mice had body weight increase rate of 1.9, 1.5, 1.4, 1.3 and 1.1%/day, with correlation coefficients of 0.796, 0.952, 0.967, 0.961 and 0.977, respectively.

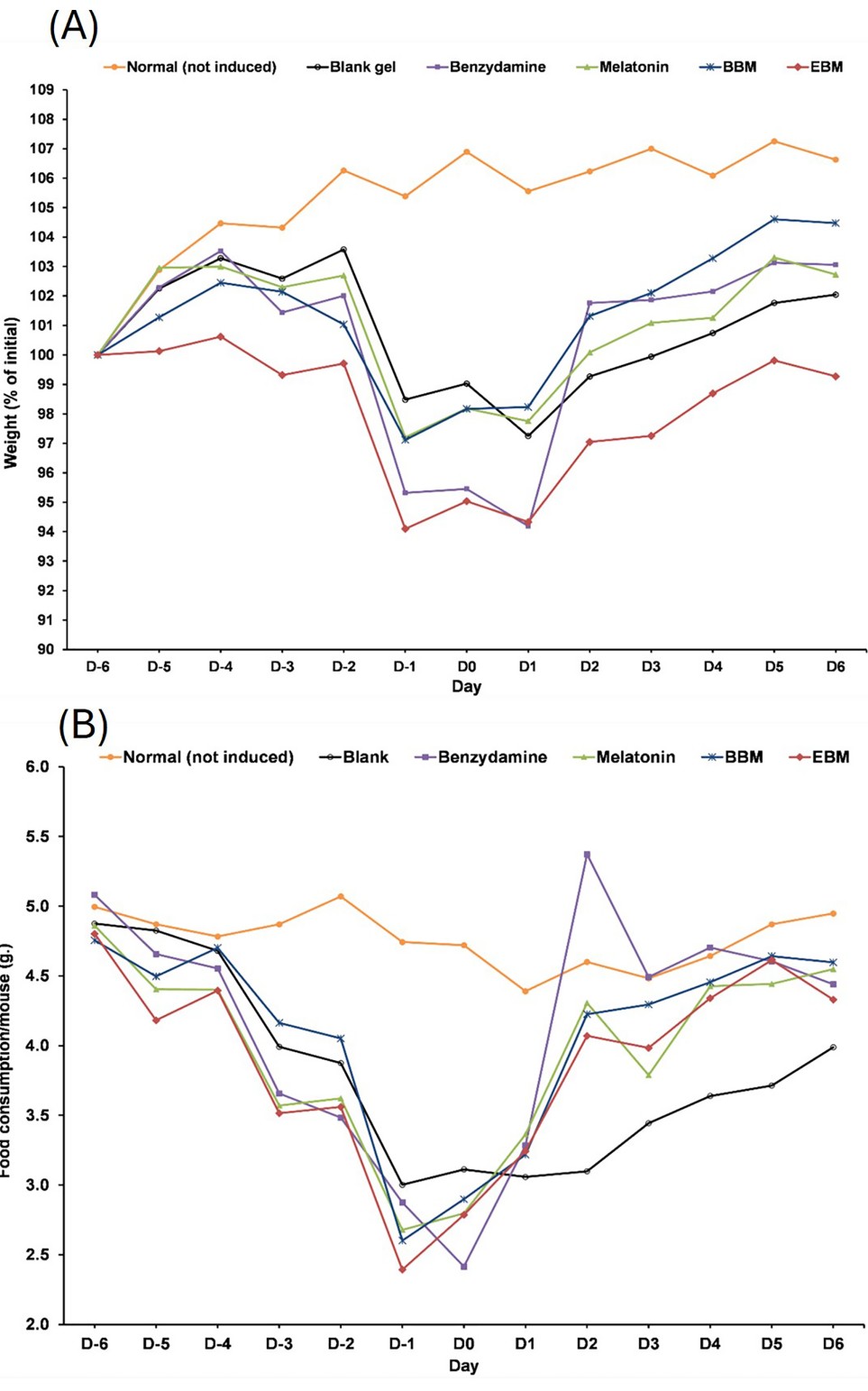

**Figure 6 (A) Body weight of mice during the 5-FU-induced oral mucositis experiment. Weight was calculated as a relative to the initial weight. (B) Food consumption of mice in each group during the 5-FU-induced oral mucositis experiment.** The amount of food consumption was calculated per mice ($n = 6$).

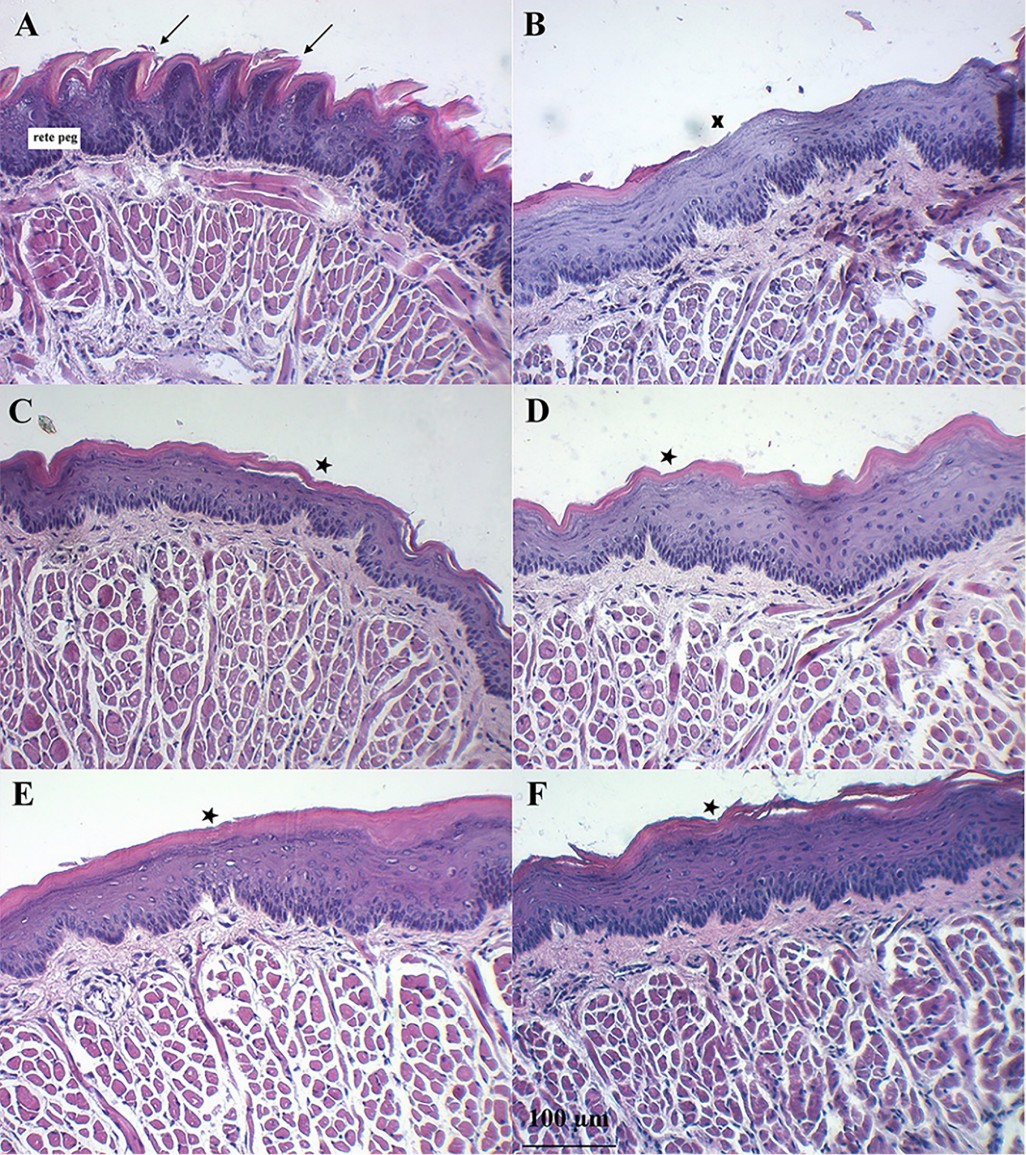

**Figure 7 Histology evaluation by H&E stained of a tongue of mice after the end of the 5-FU induced oral mucositis experiment (20 × magnification).** (A) Normal mice, (B) 5-FU with blank gel. (C) 5-FU with benzydamine gel, (D) 5-FU with melatonin gel, (E) 5-FU with BBM gel, and (F) 5-FU with EBM gel. Note: Arrows = filiform papillae were a cone-shaped structure, Star = small separation of the covering keratinized epithelium, X = loss of keratinized epithelium.

Figure 6B shows gradual reductions in daily weights of food consumed by 5-FU-treated mice, compared to the steady weights of about 5 g per mouse (4.77 ± 0.19). The weights of food consumed gradually increased from D1 in all treatment groups, except the placebo-treated group, which were significantly lower than the other 5-FU-treated groups on D3, D4, D5 and D6 ($p < 0.05$).

Figure 7 shows the dorsal surface of a normal mouse tongue covered by keratinized epithelium with an underlying basement membrane and connective tissues. The filiform papillae were a cone-shaped structure (Fig. 7A, arrows). The examination of sections of

benzydamine, melatonin, BBM, and EBM groups revealed the preservation of tissue integrity. They exhibited a small separation of the covering keratinized epithelium (star) and decreased rete pegs (Figs. 7C–7F). Examination of the tissues sections of the blank group revealed the loss of keratinized epithelium (X). Some surfaces were detached and lacked integrity. There was a separation between muscle fibers (Fig. 7B). These changes were probably due to the oxidative stress induced by the chemotherapeutic agent, 5-FU. The chemotherapeutic agent produced ROS, which can injure several macromolecules in the cell such as proteins, DNA and lipids (Altayeb & Salem, 2017).

## DISCUSSION

The results from this study revealed that melatonin, BBM and EBM exhibited different profiles in cytoprotective effects against $H_2O_2$ (Fig. 3B). Melatonin could not protect against cell damage at low concentrations (i.e., <50 μM) but showed significant protection with an increasing trend at concentration ≥50 μM ($p < 0.05$). The maximum and significant protective effect of about 65% was observed with the lowest concentration (12.5 μM) of BBM ($p < 0.001$). EBM at concentrations of 25 to100 μM provided about 30 to 40% cytoprotection. In a previous study, melatonin at 0.8 mM could protect keratinocytes from $H_2O_2$-induced cytotoxicity (She et al., 2014). Additionally, melatonin has been reported to exert cytoprotection via the autograph pathway (Lee et al., 2016), and substitution at the N1-position could protect human gingival fibroblasts from $H_2O_2$-induced toxicity (Phiphatwatcharaded et al., 2017). BBM, a hydrophobic melatonin derivative with a 4-bromobenzoyl substitution at N1-position, and EBM, another derivative with a halogenated aromatic substitution at the N-amide side chain or N2-position (Panyatip, Nunthaboot & Puthongking, 2020), enhanced the antioxidant activities of melatonin, similar to the melatonin derivative with a halogenated aromatic side chain substitution at N2-position (Shirinzadeh et al., 2010). Our results comparatively indicate that the cytoprotective effects of both derivatives were enhanced compared to those of melatonin hydrophobic substituted derivatives.

Although the melatonin derivatives (BBM and EBM) showed no toxicity to cells in the cytotoxicity experiment which was performed in the presence of FBS in culture medium, a decreasing trend in the percentage of protective effects of BBM on $H_2O_2$-damaged cells was observed as concentrations increased. In addition, $H_2O_2$ increased the susceptibility of the cells to oxidative stress, suggesting that BBM may also be toxic to cells at high concentrations under poor nutritional conditions, as this experiment was performed in serum-free medium. Abramowicz et al. (2018) demonstrated that the absence of FBS in cell culture medium resulted in the inhibition of cell viability in seven cell lines. This may further decrease cell viability in the $H_2O_2$-treated condition. At 50 μM, the melatonin derivative BBM exhibited the highest protective effect against $H_2O_2$-induced cell damage, followed by EBM and melatonin, respectively, despite some cytotoxic effects caused by BBM at this concentration, thus it was used in the comparative studies.

Although 5-FU, a chemotherapeutic agent, is involved in the generation of ROS and up-regulation of proinflammatory cytokines, resulting in oral mucositis, the cytotoxicity of 5-FU-damaged HaCaT cells was not as obvious as that of $H_2O_2$, with 0.02 mM of 5-FU

giving 69% viability. This value is consistent with other reports using human oral cancer cells (*Takano et al., 2015*), endothelial cells, and cardiac myocytes. *Focaccetti et al. (2015)* and *dos Santos Filho et al. (2018)* reported that the cell viability of HaCaT cell was significantly reduced in a concentration-dependent manner with 5-FU (0.078–10 μg/ml), and they choose the concentration that caused 33% cell viability for their chemoprotective study. *Harada et al. (2018)* studied the effect of compounds on HaCaT cell proliferation in 5-FU (2 μg/ml) induced cell damage. Our chemoprotective studies indicated the highest effect of BBM of about 30–40%, followed by EBM. The hormesis, observed as U-shaped dose-viability curves of BBM and EBM, not melatonin, with magnitudes of about 16–20% within a range of 0.001–12.5 μM, were affected by 5-FU co-cultured cells, suggesting an overload on cell responses. BBM and EBM which were co-cultured with 5-FU, were sub-therapeutic at nanomolar levels (0.001–0.1 μM) and therapeutic at micromolar levels (1–12.5 μM). Melatonin did not show a U-shaped phenomenon. Melatonin and BBM showed slight decreases in viabilities, but not reaching the toxic range, at concentrations >12.5 μM. EBM, however, showed sharp decreases in viabilities as concentrations were >12.5 μM, suggesting toxic potential at high doses.

Cellular antioxidant activity was tested based on cellular esterase function using DCFH-DA, a fluorescence probe dye cleaved into DCFH, which was hypothesized to be inhibited by the intracellular test compounds. AAPH was the reagent used to generate peroxyl (ROO•) radicals that oxidize DCFH to fluorescent DCF. BBM and EBM were previously shown to enhance scavenging activities of peroxyl radicals using Oxygen Radical Absorbance Capacity (ORAC) assays (*Panyatip, Nunthaboot & Puthongking, 2020*) while enhanced cellular antioxidant activities of 2 melatonin derivatives at 50 μM with *N1*-substituted melatonin in fibroblasts were also reported using AAPH (*Phiphatwatcharaded et al., 2017*). However, the results of this study illustrated that BBM and EBM could not enhance the cellular antioxidant activity from their parent compound. Mitochondria are one of the important organelles and a major source of the production of energy from oxygen as well as ROS and RNS (*Yang et al., 2017*). To clarify the potential of melatonin derivatives to protect against mitochondria dysfunction, experiments evaluating mitochondria oxidative stress need to be further studied. Intracellular ROS levels of 5-FU-damaged HaCaT cells were also significantly and concentration-dependently suppressed by pretreatment with 12.5, 25, 50, 100 μM of melatonin, BBM or EBM. Both melatonin derivatives could also enhance ROS reduction to lower levels than melatonin. This finding complemented the chemoprotective results, suggesting that BBM and EBM gave higher chemoprotective effects than melatonin ($p < 0.05$), possibly due to ROS suppression and other factors. Chemotherapy induces epithelial cell injury through various mechanisms, some of which are mediated by the generation of ROS (*Sonis et al., 2004*). ROS can also activate the NF-κB and p53 pathways, leading to the production of pro-inflammatory cytokines such as TNF-α, IL-1β, IL-6 (*Oronsky et al., 2018*). 5-FU is well known to induce cell apoptosis due to the production of ROS and mitochondria membrane depolarization (*Zorov, Juhaszova & Sollott, 2006*). *dos Santos Filho et al. (2018)* reported that HaCaT cells exposed to 5-FU showed increased pro-inflammatory cytokines, especially TNF, IL-1β, IL-6, and IL-8. In the present study, the results demonstrated that melatonin and its

derivatives could suppress 5-FU-induced generation of ROS. Moreover, melatonin and its derivatives exhibited anti-inflammatory activities by inhibiting nitric oxide, TNF-α, and IL-6 levels, suggesting that melatonin derivatives may inhibit NF-κB in cells. Therefore, melatonin and its derivatives could prevent mucositis pathobiology due to their antioxidant and anti-inflammatory activities.

Symptoms indicating mucositis in 5-FU-treated mice include diarrhea, losses of hair, body weight loss, and decreased appetite. However, the appearance of the oral cavity of mice did not clearly indicate mucositis. The average body weights of 5-FU + EBM-treated mice on D2, D3, D4, D5 and D6 were significantly higher than those of 5-FU-treated mice ($p < 0.05$). The average body weights of BM-treated mice were insignificantly different from those of the benzydamine and melatonin-treated groups ($p > 0.05$). The least food consumption and the lowest body weights were found in 5-FU/placebo-treated mice on D2, D3, D4, D5 and D6 ($p < 0.05$, all), suggesting that there could be potential benefits for topical preparations containing melatonin, BBM, EBM or benzydamine in oral mucositis. Histological examination of H&E stained dorsal tongues taken from representative 5-FU-treated mice after being treated for 5 days with gels of placebo, benzydamine, melatonin, BBM, EBM (Figs. 7B–7F, respectively) were compared with normal mice, Fig. 7A. Keratinized epithelium and cone-shaped filiform papillae with an underlying basement membrane and connective tissues were histologically characteristic of the dorsal tongue. Benzydamine, melatonin, BBM and EBM, obviously provided integrity to the outer layer. The thickness of the epithelial layers with dense cells and intact nuclei, that were observed with melatonin, BBM and EBM-treated mice, indicated healing potential.

The pathobiology of mucositis is a complex process. There is limited information about the events leading to mucosal damage during cancer treatment. Pro-inflammatory cytokines play an important role in the pathogenesis of mucositis. Our results showed that melatonin or melatonin derivatives; BBM and EBM might be helpful in oral mucositis to relieve symptoms. The possible mechanism is that melatonin and its derivatives could reduce pro-inflammatory cytokines and inhibit croton oil-induced ear edema (*Sangchart et al., 2021*). Moreover, melatonin and its derivatives have free radical scavenging and antioxidant properties enabling them to protect the oral mucosa from oxidative damage. In addition, melatonin and its derivatives increased cell proliferation, which could be helpful to reduce the severity of mucositis.

## CONCLUSIONS

The melatonin derivatives BBM and EBM might be helpful to relieve oral mucositis symptoms by mechanism involving cellular antioxidant properties that protect the oral mucosa from oxidative damage and could reduce the severity of mucositis pathobiology.

## ACKNOWLEDGEMENTS

The authors thank Glenn Borlace for English language editing through the KKU Publication clinic.

### Funding

The research has received funding support from Khon Kaen University-National Research Council of Thailand (KKU-NRCT), and the National Science, Research and Innovation Fund (NSRF), Thailand *via* the Fundamental Fund of Khon Kaen University. The funders had no role in study design, data collection and analysis, decision to publish, or preparation of the manuscript.

### Grant Disclosures

The following grant information was disclosed by the authors:
Khon Kaen University-National Research Council of Thailand (KKU-NRCT).
National Science, Research and Innovation Fund (NSRF), Thailand *via* the Fundamental Fund (FF) of Khon Kaen University.

### Competing Interests

The authors declare that they have no competing interests.

### Author Contributions

- Pramote Mahakunakorn conceived and designed the experiments, performed the experiments, analyzed the data, prepared figures and/or tables, authored or reviewed drafts of the article, and approved the final draft.
- Pimpichaya Sangchart performed the experiments, analyzed the data, prepared figures and/or tables, authored or reviewed drafts of the article, and approved the final draft.
- Panyada Panyatip performed the experiments, prepared figures and/or tables, and approved the final draft.
- Juthamat Ratha performed the experiments, prepared figures and/or tables, authored or reviewed drafts of the article, and approved the final draft.
- Teerasak Damrongrungruang conceived and designed the experiments, analyzed the data, authored or reviewed drafts of the article, and approved the final draft.
- Aroonsri Priprem conceived and designed the experiments, analyzed the data, authored or reviewed drafts of the article, and approved the final draft.
- Ploenthip Puthongking conceived and designed the experiments, performed the experiments, analyzed the data, prepared figures and/or tables, authored or reviewed drafts of the article, and approved the final draft.

### Animal Ethics

The following information was supplied relating to ethical approvals (*i.e.*, approving body and any reference numbers):

The Northeast Laboratory Animal Center, Khon Kaen University, Thailand.

### Data Availability

The raw data are available in the Supplemental Files.

## Supplemental Information

Supplemental information for this article can be found online at http://dx.doi.org/10.7717/peerj.17608#supplemental-information.

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
