# Peer review of "In vitro cytoprotective and in vivo anti-oral mucositis effects of melatonin and its derivatives"

_PeerJ, doi:10.7717/peerj.17608_

## Round 0.1 · original submission · Major Revisions

Overall, the reviewers recommend major revisions. The manuscript advances knowledge of treatments for oral mucositis and provides comprehensive data measured both in vitro and in vivo. Please provide responses to each comment posed by reviewers and indicate where in the manuscript any revisions are being made. In addition to the reviewer comments, also please address the following:

1. Authors used human keratinocyte (HaCaT) cell line that relates to skin. How is this related to oral mucositis? Please add explanation in manuscript.

2. Figure captions don’t always match figures. Many captions mention several graphs (A, B) but they are missing. In the text there are references to Figures 1-7 but the figures themselves are labeled 1-11.

3. Results lines 248-252: show this data in supplementary info.

4. Methods lines 227-231: show the control for blank gel that is missing.

5. In histology images add labels for the observations described in lines 324-330

**Language Note:** The review process has identified that the English language must be improved. PeerJ can provide language editing services - please contact us at [email protected] for pricing (be sure to provide your manuscript number and title). Alternatively, you should make your own arrangements to improve the language quality and provide details in your response letter. – PeerJ Staff

Reviewer 1 ·

Basic reporting

Introduction: not only 5-FU induces mucositis, other chemotherapeutic drugs with this side effect should be indicated
Materials & Methods. The immortalized human keratinocyte (HaCaT) cell line (American Type Culture Collection, 124 Manassas, VA, USA): verify that the cells were provided by ATCC.
Figure 3: In the figure legend is indicated that the control bar represents H2O2. If this is correct, H2O2 gives the 100% of viability? And the treatments reduce the cell viability respect to H2O2?
The same doubts for Figure 5
Figure 11: Histological differences should be indicated in the legend and arrows should be used in the figureto indicate the details of the histological differences between control and treatments
The results section should be organized in paragraph with titles referring to the results not to the number of the figures
The discussion section should be reduced in the first part

Experimental design

The experimental design is, in principle, correct but more evaluation by the in vivo experiments (e.g. parameters of inflammation) could add a significant value to the research.

Validity of the findings

No comment

Reviewer 2 ·

Basic reporting

Some parts of the manuscript, particularly in the introduction and discussion, could use further refinement for clarity and brevity. Consider revising lengthy sentences to improve readability. Engaging a colleague proficient in English or a professional editing service could be beneficial.

Experimental design

1. The section on materials and methods is well-detailed yet could be enhanced by elaborating on the statistical analysis, especially regarding adjustments for multiple comparisons. A clearer explanation of the statistical methods, including any specific corrections (e.g., Bonferroni, Sidak), would enhance reproducibility.
2. It'd be really helpful to spell out what's going on with the control groups a bit more clearly. Making sure readers get the difference between your experiment setups and the controls is super important for the clarity.

3. Throwing in some explanation about why you picked the number of samples you did would be great. Also, a bit about how you've made sure your experiments can be repeated by others would add a lot to the trustworthiness of

Validity of the findings

Although the results are clearly presented, a more robust discussion on the significance of these findings in relation to existing studies could fortify the manuscript. A direct comparison between the effects of BBM and EBM with those of melatonin on oral mucositis, in light of previous research, would offer a more comprehensive understanding of their relative efficacy.

Additional comments

The visual elements like figures and tables are well-crafted but some can be optimized for better clarity. Ensuring high-resolution figures and readable labels, even when printed in black and white, would improve their effectiveness.

·

Basic reporting

The study investigates the effects of melatonin and its N-amide derivatives, specifically BBM and EBM, on oral mucositis induced by chemotherapy, particularly 5-Fluorouracil (5-FU). The motivation behind the study is rooted in addressing chemotherapy-induced toxicity, with a focus on exploring potential therapeutic agents that can mitigate such side effects.

The article contains numerous grammatical mistakes that require correction. I suggest using online grammar checking tools, such as the ones I have listed as examples.

Original: "In this study purposed therefore to evaluate the eûect of melatonin..."
Corrected: "This study aimed to evaluate the effect of melatonin..."

Original: "Moreover, the weight of mice and food consumption recovered initially faster in BBM group."
Corrected: "Moreover, the weight of mice and food consumption recovered more quickly in the BBM group."

Original: "Melatonin is an endogenous hormone which mainly produced and secreted by a pineal gland..."
Corrected: "Melatonin is an endogenous hormone that is mainly produced and secreted by the pineal gland..."

Ensure the correct and consistent use of biological and chemical terminology, including capitalization and italicization where appropriate (e.g., species names should be italicized).

The first time an abbreviation or acronym is used, it should be defined (e.g., ROS for Reactive Oxygen Species). However, the authors do not need to define it repeatedly.

Experimental design

The research design encompasses both in vitro and in vivo experiments. The in vitro part examines the cytoprotective effects of melatonin, BBM, and EBM against H2O2 and 5-FU-induced cytotoxicity in human keratinocyte (HaCaT) cells. The in vivo component assesses the efficacy of these compounds in preventing oral mucositis in mice treated with 5-FU. This dual approach aims to provide comprehensive insights into the potential therapeutic benefits of the compounds studied.

The manuscript lacks sufficient transition from figure to figure and rationale behind each experiment or measurement. Please ensure that more transitions and explanations are included.

Validity of the findings

The results indicate that BBM offers the highest level of protection against H2O2-induced damage in HaCaT cells, followed by EBM and melatonin. Similar protective trends were observed against 5-FU-induced cytotoxicity. In vivo, the study reports that gel formulations containing melatonin, BBM, and EBM protected against tissue damage from 5-FU in mice, comparable to the standard compound benzydamine. The findings suggest the potential of BBM and EBM as new therapeutic agents for oral mucositis treatment.

Specifically, I disagree with the authors' claim regarding the cytotoxicity of 5-FU-damaged HaCaT cells determined by the MTT assay. Figure 4A shows significant cell viabilities ranging between 62-76% within the investigated concentration range of 270, indicating a concentration-dependent manner. However, the term "concentration-dependent manner" lacks clarity, and the observed trend contradicts the expected concentration effect, as cell viability is greater with 0.15 5-FU than with 0.08 5-FU.

Additional comments

Please add more information and explanation to Figures 1 and 2.

For Figure 3, please label the x-axis with H2O2 concentration.

In the legend of Figure 3, please elaborate on the significance denoted by #p < 0.05 versus melatonin. This symbol appears only in Figure 6; therefore, please delete it from the legend for figures where it is not present.

Figure 4 in the document corresponds to Figure 3B in the text and legend. Please ensure that each figure is correctly labeled(applies to every figure after).

---

## Round 0.2 · accepted · Accept

All appropriate changes were made by authors.

Reviewer 2 ·

Basic reporting

none

Experimental design

none

Validity of the findings

none

·

Basic reporting

I have no further comments.

Experimental design

I have no further comments.

Validity of the findings

I have no further comments.

Additional comments

I have no further comments.